# EpilepsyFM: Foundation Model for Learning Generalized Epileptic Representations from EEG and SEEG Signals

## Abstract

Extracranial electroencephalography (EEG) and intracranial stereoelectroen-cephalography (SEEG) are crucial for epilepsy diagnosis. However, existing deep learning models often limit themselves to specific signal types and application scenarios, leading to challenges in generalization and perception capabilities. While large language models excel in natural language processing, they cannot effectively capture the disease-specific signal features in the highly specialized field of epilepsy, and the lack of pre-training data restricts their generalization ability. To address these issues, we propose a Epilepsy Foundation Model (EpilepsyFM), a domain-specific foundational model that considers the mechanisms of seizure and propagation in epilepsy. EpilepsyFM learns a generalized representation of epilepsy through unsupervised pre-training across various signal types, data formats, and sources, and optimizes multiple epilepsy-related downstream tasks through fine-tuning. We collected clinical EEG and SEEG data from multiple patients at a first-class hospital, as well as the currently largest publicly available epilepsy dataset, the TUH series, ensuring diversity in representation learning. First, the neural activity signals are segmented into multiple patches, and a discrete EEG and SEEG neural tokenizer is trained to construct a domain-specific neural codebook for epilepsy. Then, EpilepsyFM takes into account the mechanisms of clustered neuronal discharges in epilepsy and designs a channel set masking strategy to enhance the model's ability to capture the spatiotemporal character-istics of the signals. The model fully utilizes the multi-dimensional propagation characteristics of seizures through temporal, spectral, and spatial encoder modules, achieving comprehensive representation of complex neural signals. Extensive experiments show that EpilepsyFM achieves state-of-the-art performance in a vari-ety of domain-specific tasks, including seizure detection and both short-term and long-term predictions of neural signals, demonstrating strong generalization ability and broad clinical application potential.

## 1 Introduction

Epilepsy is a chronic neurological disorder caused by abnormal discharges of brain neurons, affecting approximately 70 million people worldwide, making it one of the most common neurological diseases (Asadi-Pooya et al., 2023; Specchio et al., 2022). Its pathological mechanisms are complex and varied, with diverse clinical manifestations. Persistent epileptic seizures not only endanger patients' health but also place a heavy burden on medical resources (Kanner & Bicchi, 2022). Extracranial electroencephalography (EEG) and intracranial stereoelectroencephalography (SEEG) are effective neurosurgical methods for recording brain activity in epilepsy patients during ictal and interictal periods, with intracranial SEEG widely regarded as the gold standard for diagnosing epileptic seizures (Supriya et al., 2021; Narasimhan et al., 2020; Yan et al., 2024).

Currently, deep learning models for epilepsy are typically designed for specific neural signals and application scenarios (Shoeibi et al., 2021; Usman et al., 2021; Peh et al., 2023). Although there has been some progress in seizure detection and prediction using EEG, such as using graph convolutional networks (Li et al., 2020) and generative adversarial networks (Rasheed et al., 2021), two core issues remain: first, clinical epilepsy EEG and SEEG are scarce, particularly the limited availability of

SEEG as the gold standard, which prevents many models from being thoroughly validated in clinical applications; second, most existing models are designed for specific tasks, performing exceptionally well on single tasks but lacking generalization capability in real clinical settings.

In recent years, the rise of general-purpose large language models (LLMs) (Vaswani, 2017; Thirunavukarasu et al., 2023; Yuan et al., 2024a), such as ChatGPT, has demonstrated their powerful capabilities in addressing multi-task and multi-domain problems. Compared to traditional deep learning models, LLMs effectively overcome the limitations of single-task models through pre-training and fine-tuning. In the field of medical information, LLMs provide intelligent support for disease diagnosis by processing vast datasets, significantly improving healthcare efficiency (Li et al., 2023; Thirunavukarasu et al., 2023). Recent studies have introduced various foundational models for analyzing brain neural activity, showcasing exceptional performance. For example, Cui et al. (Cui et al., 2024) proposed Neuro-GPT, which combines an EEG encoder with a GPT architecture to address challenges related to the scarcity and heterogeneity of EEG data. Yi et al. (Yi et al., 2023) aimed to bridge gaps between different EEG resources, introducing the MMM framework for cross-dataset pre-training, while Jiang et al. (Jiang et al., 2024) presented a unified model that successfully navigates issues such as mismatched EEG electrode counts.

Although general-purpose LLMs perform exceptionally well, domain-specific models often demonstrate superior performance in specialized fields (Pal et al., 2024; Zhang et al., 2024; Arefeen et al., 2024). These models are optimized to capture unique features relevant to their domains. For example, Gu et al. (Gu et al., 2021) highlighted the advantages of training domain-specific language models from scratch in biomedical natural language processing, while Xie et al. (Xie et al., 2023) proposed a cost-effective strategy that successfully built domain-specific models, achieving significant improvements in financial tasks. For epilepsy, a clinical neurological disorder, we hope to break through the problem that deep learning can only solve a single task, and avoid the limitation of relying only on LLMs related to generalized brain activity to deal with various data sources (e.g., motor imagery, sleep detection, etc.). Our goal is to design a clinical-focused foundation model that concentrates on epilepsy. This model will learn the generalized representations of intra- and extracranial signals associated with epilepsy, aiming to effectively tackle various tasks related to the condition. However, we still face several challenges in this process:

**1) Lack of sufficient EEG and SEEG data for epilepsy:** Although several publicly available EEG datasets for epilepsy exist, SEEG data remains extremely scarce. Collecting EEG data poses significant challenges due to individual differences in seizures and the considerable time and effort required for expert annotation.

**2) Lack of consideration for the mechanisms of seizure:** The mechanisms of epilepsy are complex and involve abnormal discharges from various brain neurons. Improving the model's understanding of the dynamic changes during seizure processes remains a critical challenge.

**3) Lack of consideration for the propagation process of epilepsy:** After a seizure, the propagation process spreads over time to different brain regions, accompanied by low-amplitude, high-frequency oscillations. This process involves information across time, space, and frequency. Capturing signal changes during this propagation and learning the features of EEG activity related to epilepsy are key to building an efficient model.

To address aforementioned challenges, our goal is to design a domain-specific foundational model for epilepsy, termed EpilepsyFM, which considers seizure onset and its propagation mechanisms, effectively handling EEG and SEEG data while achieving excellent performance across different epilepsy task states. We first segment neural activity signals into multiple patches and, prior to pre-training, leverage the concept of VQ-VAE (Van Den Oord et al., 2017) to train a discrete neural tokenizer, thereby constructing a neural vocabulary specific to the epilepsy domain. Subsequently, during the pre-training phase, we design a channel set masking strategy that reflects the characteristics of clustered epileptic discharges, randomly masking sets of neural patches and predicting the masked patches from visible ones, enhancing the model's learning capability and accelerating the training speed through a symmetric masking strategy. Additionally, EpilepsyFM incorporates temporal, spatial, and spectral encoding methods aimed at capturing the dynamic features during the propagation process of epilepsy. The main contributions of this work are summarized as follows:

**1) Robust data foundation:** We collected clinical EEG and SEEG epilepsy data from a a first-class hospital's neurosurgery department and combined it with the largest publicly available epilepsy

EEG dataset, the TUH series, to serve as the pre-training foundation for EpilepsyFM. The labels for downstream tasks are meticulously annotated by two specialized physicians from the hospital.

**2) Channel set masking pre-training strategy:** In response to the clustered discharge mechanisms observed during epileptic seizures, we designed a self-supervised pre-training strategy utilizing localized channel set masking. This approach enables more accurate capture of neural activity both within and across channels while improving computational efficiency through symmetric masking.

**3) Temporal-spatial-spectral encoding:** To gain deeper insights into the propagation mechanisms of epilepsy, we developed temporal, spatial, and spectral encoding methods. These techniques dynamically capture the features of neural activity associated with epilepsy, providing a comprehensive understanding of its complex dynamics.

**4) Diverse downstream tasks:** We conducted comprehensive experimental studies on several key tasks related to epilepsy, including seizure detection, as well as short-term and long-term signal prediction. The results indicate EpilepsyFM's adaptability across different tasks, highlighting its broad potential for clinical applications.

## 2 METHOD

The overall architecture of EpilepsyFM is depicted in Figure 1. In this architecture, EEG and SEEG signals with varying channel counts and temporal lengths are input into the model to learn generalized representations of epileptic neural activity, which are then encoded for downstream tasks.

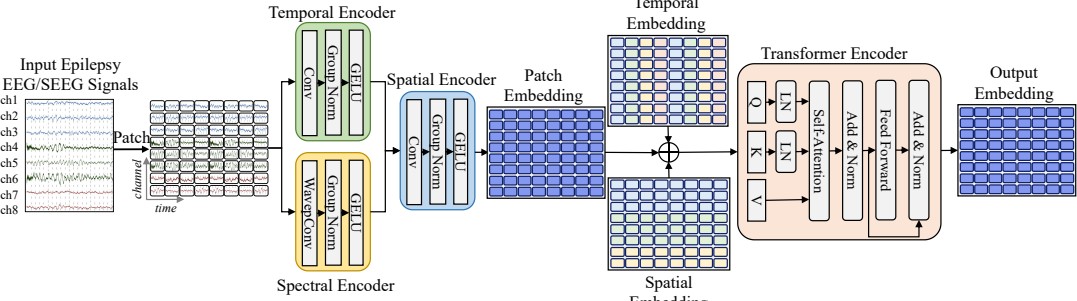

Figure 1: The overall architecture of EpilepsyFM, the Epileptic Neural Transformer. EEG or SEEG signals are first segmented into fixed-size patches. A temporal encoder and a spectral encoder extract time and frequency features from each patch, which are then concatenated. The spatial encoder captures channel information and integrates the temporal and spatial embeddings into patch embeddings. Finally, these embeddings are fed into the transformer encoder in patch order to produce the final output.

### 2.1 DEFINITIONS AND NOTATIONS

The large publicly available epilepsy EEG datasets TUEP, TUSL, and TUSZ, along with the collected clinical long-term EEG and SEEG datasets (XJSZ-EEG and XJSZ-SEEG), are utilized as pre-training data (for more details, see Appendix B). The multi-channel EEG and SEEG signals are represented as $X \in \mathbb{R}^{C \times P}$, where $C$ represents the number of EEG and SEEG channels, and $P$ represents the number of sampling points with a sampling rate of $S$. The set of channels used in EEG and SEEG is defined as $C_X = \{c_{i1}, c_{i2}, \ldots, c_{iC}\}$ for EEG and $C'_X = \{c_1, c_2, \ldots, c_{C'}\}$ for SEEG, where $C_X \subseteq C_{EEG}$ and $C'_X \subseteq C_{SEEG}$, with $C_{EEG}$ being the complete set of channels defined by the international 10-20 system (Jurcak et al., 2007), and $C_{SEEG}$ representing the set of all SEEG channels actually used in clinical experiments.

### 2.2 MODEL ARCHITECTURE

We introduce a general architecture for learning epileptic neural activity — Epileptic Neural Transformer, which can handle scalp EEG signals and intracranial SEEG signals with different data formats,

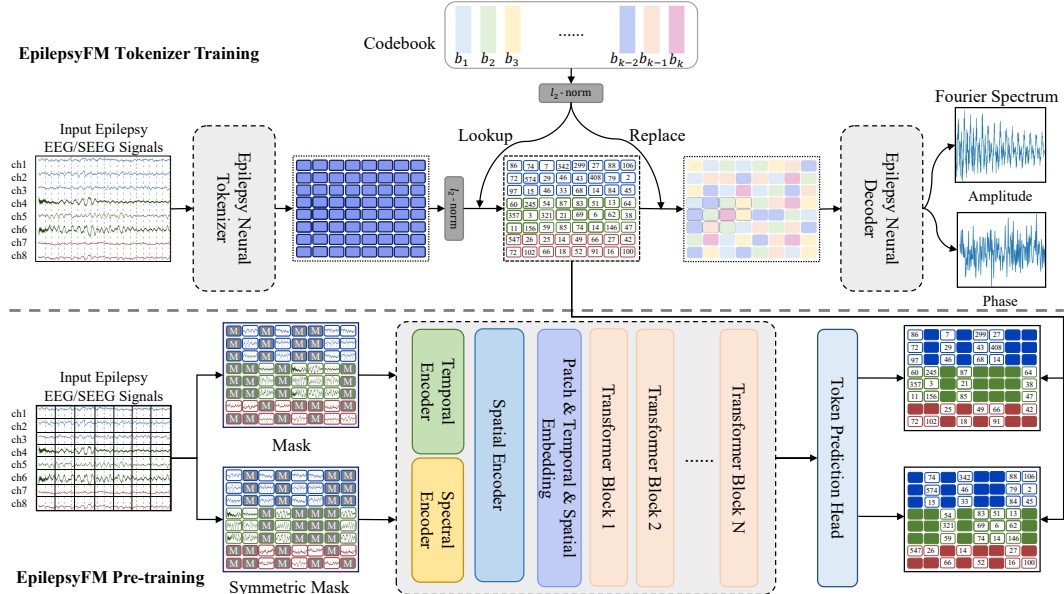

Figure 2: Training of Epilepsy EEG&SEEG Tokenizers and EpilepsyFM Pre-training. **(a) EpilepsyFM Tokenizer training**: The tokenizer discretizes epileptic EEG and SEEG signals into discrete neural tokens by decoding epileptic neural activity. **(b) EpilepsyFM Pre-training**: During pre-training, patches of epileptic EEG and SEEG are masked according to the channel set, with the objective of predicting the masked neural tokens from the unmasked patches.

channel numbers, and sampling points. The model framework is illustrated in Figure 1. For each sample $x$, we divide it into $N$ patches using a non-overlapping time window $W$ (Bao et al., 2021), resulting in $x = \{x_{c_{ij},k} \in \mathbb{R}^{C \times W} | j = 1, 2, \ldots, C, k = 1, 2, \ldots, \lfloor \frac{P}{W} \rfloor\}$, where $N = C \lfloor \frac{P}{W} \rfloor$.

**Temporal Encoder.** Due to the abnormal temporal variations in EEG and SEEG signals during epileptic seizures, we propose a temporal encoder to learn temporal features before the self-attention mechanism interacts between patches. The encoder consists of multiple temporal convolution blocks, each composed of a convolution layer, a group normalization layer, and a GELU activation function (Hendrycks & Gimpel, 2016). The output of the temporal encoder can be expressed as:

$$\varepsilon_t = \left\{ e^t_{c_{ij},k} \in \mathbb{R}^d | j = 1, 2, \ldots, C, k = 1, 2, \ldots, \lfloor \frac{P}{W} \rfloor \right\} \tag{1}$$

where $d$ is the dimension of the embeddings.

**Spectral Encoder.** Due to epileptic ictal or interictal periods, patients' brain activity signals exhibit characteristic waves such as spikes, sharp waves, and sharp-slow waves, which are high-frequency, low-amplitude oscillations, all accompanied by changes in frequency. It is therefore necessary to analyze the signal variations at different frequency components during epileptic ictal or interictal periods. To this end, we propose a wavelet packet convolution (WavepConv) to extract frequency features from EEG and SEEG signals (Wang et al., 2021). In the aforementioned approach, the input to the spectral encoder can be expressed as $e^f_i$, where:

$$\varepsilon_w = e^f_i(N - \frac{R}{2} + 1), \ldots, e^f_i(N-1) \copyright e^f_i(0), \tag{2}$$
$$\ldots, e^f_i(N-1) \copyright e^f_i(0), \ldots, e^f_i(\frac{R}{2} - 2)$$

$$y_A(t) = \sum_{r=0}^{R} \varepsilon_w(s \times i - r) \times g(r)$$
$$y_D(t) = \sum_{r=0}^{R} \varepsilon_w(s \times i - r) \times h(r) \tag{3}$$

where, $\varepsilon_w$ represents the output of the periodic padding. Periodic padding ensures the minimum length of wavelet packet decomposition, preserving the original characteristics of the input signals and avoiding boundary effects. $\copyright$ represents the concatenating operation. $R$, $s$ represent the size of the

convolution kernel and stride, respectively. $y_A(t)$ and $y_D(t)$ represent the approximate coefficients and fine coefficients, respectively. And $g$, $h$ are a pair of convolution kernels (Shi et al., 2020).

**Spatial Encoder.** Due to the characteristic of abnormal discharge signals spreading from the epileptic focus to other brain regions over time during a seizure, we propose a spatial encoder to effectively capture this spatial diffusion pattern (Zhang et al., 2019). The spatial encoder is designed to learn spatial features both within and across channels in EEG and SEEG signals. This encoder is composed of multiple spatial convolution blocks, similar to the temporal encoder, but with a greater focus on capturing the spatial correlations between channels. We concatenate the features encoded in both time and frequency from EEG and SEEG signals ($\varepsilon p = \varepsilon_t \copyright \varepsilon_w$), and the output patch embedding of the spatial encoder can be expressed as:

$$\varepsilon_p = \left\{ e^p_{c_{ij},k} \in \mathbb{R}^d | j = 1, 2, \ldots, C, k = 1, 2, \ldots, \lfloor \frac{P}{W} \rfloor \right\} \tag{4}$$

where $d$ is the dimension of the embeddings.

**Temporal & Spatial Embedding.** To enable the model to capture both temporal and spatial information of patch embeddings (Petukhova et al., 2024), we constructed two learnable embedding lists: temporal embeddings $te = \{te_1, te_2, \ldots, te_{t_{\max}}\}$ and spatial embeddings $se = \{se_1, se_2, \ldots, se_C\}$, both of which have a dimension of $d$. In this process, the hyperparameter $t_{\max}$ determines the maximum number of temporal patches, satisfying the condition $t_{\max} \geq N$. Given arbitrary patch embedding $e_i$ from the spatial encoder, the result by adding the corresponding temporal and spatial embeddings can be expressed as:

$$\varepsilon_{ineb} = \{e^p_{c_{ij},k} + te_k + se_c | j = 1, \ldots, C, k = 1, 2, \ldots, \lfloor \frac{P}{W} \rfloor\} \tag{5}$$

where, $\varepsilon_{ineb}$ is the input to the Transformer encoder.

**Transformer Encoder.** The sequence of embedding $\varepsilon_{\text{ineb}}$ is input into the Transformer encoder (Vaswani, 2017), resulting in the output embedding $e_i \in \mathbb{R}^d$. To prevent excessively large values in the attention scores, we apply layer normalization to the queries and keys before the dot-product attention mechanism.

$$\text{Attention}(Q, K, V) = \text{softmax} \left( \frac{\text{LN}(Q)\text{LN}(K)^T}{\sqrt{d_{head}}} \right) V \tag{6}$$

where, LN represents layer normalization, and $d_{\text{head}}$ represents the dimension of each attention head. For downstream prediction tasks, the output embeddings are flattened and processed using classification or prediction heads.

## 2.3 NEURAL TOKENIZER TRAINING

**EpilepsyFM Neural Tokenizer.** As shown in Figure 2(a), we are inspired by LaBraM (Van Den Oord et al., 2017; Jiang et al., 2024); however, they only trained the tokenizer on EEG signals, which makes it incompatible with SEEG signals. To address the different signal sources in epilepsy (such as intracranial and extracranial signals), we define an epilepsy neural tokenizer $B = \{b_i | i = 1, ..., N_{\text{code}}\} \in \mathbb{R}^{N_{\text{code}} \times D_{\text{code}}}$, where $N_{\text{code}}$ represents the number of discrete epilepsy neural embeddings and $D_{\text{code}}$ represents the dimensionality of the embeddings. Given an epilepsy EEG or SEEG signal, the tokenizer encodes it into patches $q = \{q_i \in \mathbb{R}^{d_{code}} | i = 1, 2, \ldots, N\}$, where $N$ indicates the number of patches. Subsequently, the quantizer converts $q_i$ into embeddings in the epilepsy codebook. Finally, the nearest neighbor for each $q_i$ is found using the epilepsy codebook $B$ (Peng et al., 2022), which can be expressed as:

$$z_i = \arg \min_j \|\ell_2(q_i) - \ell_2(b_j)\|^2 \tag{7}$$

where, $\ell_2$ is the $\ell_2$-normalization, and $z_i$ is the vector after being quantized by the quantizer.

**EpilepsyFM Neural Decoder.** Due to the low signal-to-noise ratio of epilepsy EEG and SEEG signals, especially with EEG signals exhibiting strong randomness and nonstationarity, directly reconstructing these signals is highly challenging. Instead of reconstructing the raw signals, we predict the amplitude and phase from their Fourier spectrum, as these features more accurately reflect

brain activity. We apply the Discrete Fourier Transform (DFT) to EEG or SEEG segments to extract amplitude and phase information (Gao et al., 2023), followed by normalization to ensure stable model convergence. Then, using a epilepsy neural tokenizer and decoder, we train the model to predict these features using the mean squared error loss function, with the total loss defined as follows:

$$\mathcal{L}_T = \sum_{\boldsymbol{x} \in \mathcal{D}} \sum_{i=1}^{N} \|o_i^A - A_i\|_2^2 + \|o_i^\phi - \phi_i\|_2^2 + \|\boldsymbol{sg}(\ell_2(q_i)) - \ell_2(v_{b_i})\|_2^2 + \|\ell_2(q_i) - \boldsymbol{sg}(\ell_2(b_{z_i}))\|_2^2 \tag{8}$$

where, $o_i^A$ and $o_i^\phi$ represent the predicted amplitude and phase from the epilepsy neural decoder, respectively, while $A_i$ and $\phi_i$ represent the true amplitude and phase obtained from the Fourier transform of the EEG or SEEG signals, and $sg(\cdot)$ indicates the stop-gradient operation.

### 2.4 EPILEPSYFM PRE-TRAINING

**Masked EEG and SEEG Modeling.** As shown in Figure 2(b), to enhance the general representation learning of EpilepsyFM with extensive data, we propose a channel set masking strategy that specifically considers the mechanisms of epileptic seizures (Wei et al., 2022). Specifically, for the case where the number of channels is divisible by 3, we apply random masking to group the channels into sets of three, forming the mask $M = \{m_i | i = 1, ..., N\}$, where $m_i \in \{0, 1\}$ and a proportion $r$ of $m$ is set to 1. For the remaining channels, we use single-channel random masking, as shown in Figure 3. The rationale for using 3 channels is detailed in Appendix F. We replace the random mask $M$ with a learnable mask $e_M \in \mathbb{R}^d$; these masked patches are added to the temporal and spatial embeddings to serve as input to the Transformer encoder. Using the output hidden vectors, we predict the neural tokens related to EEG and SEEG. Ultimately, our objective training loss is defined as follows:

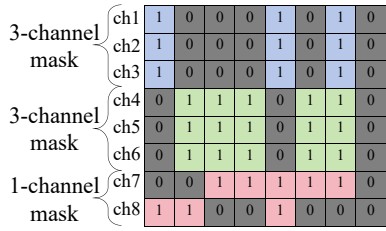

Figure 3: Epileptic EEG and SEEG electrodes are masked according to channel sets.

$$L_M = -\sum_{x \in D} \sum_{m_i = 1} \log p(b_i | e_M) \tag{9}$$

**Symmetric Masking.** Inspired by LaBraM, we propose a symmetric masking strategy to enhance training efficiency. We utilize the inverse of the mask $M$ as a new masking approach for pre-training, thereby obtaining the corresponding loss $L_{\text{sym}}^M$. Consequently, the total loss for our epilepsy base model is defined as follows:

$$L_{\text{mae}} = L_M + L_{\text{sym}}^M \tag{10}$$

## 3 EXPERIMENTS

### 3.1 DATASETS

Due to the current lack of publicly available SEEG datasets related to epilepsy, we have collected long-term clinical SEEG data from 20 patients in the neurosurgery department of a first-class hospital, as well as extensive long-term EEG data, all acquired using Neuracle equipment (i.e., XJSZ-EEG and XJSZ-SEEG). In these datasets, the EEG and SEEG signals are time-locked .

**Pre-training Datasets:** The pre-training dataset includes the largest publicly available epilepsy-related TUH EEG datasets, such as TUEP, TUSL, and TUSZ (Harati et al., 2014), as well as data from 4 patients for XJSZ-EEG and 12 patients for XJSZ-SEEG that we collected. To cover all training data and improve training efficiency, we set the time length of each sample to 4 seconds.

**Downstream Datasets:** The downstream task dataset includes TUAB, TUEV, (Harati et al., 2014) CHB-MIT (Shoeb, 2009), and data from 8 patients with XJSZ-EEG and XJSZ-SEEG. The tasks across these datasets vary. TUAB is segmented into 10-second samples, TUEV into 5-second samples, and CHB-MIT, XJSZ-EEG, and XJSZ-SEEG into 2-second samples.

For more detailed information about the data, please refer to Appendix B.

## 3.2 IMPLEMENTATION DETAILS

**Preprocessing:** We downsampled the SEEG signals from 1000Hz to 200Hz and similarly downsampled the EEG signals from various datasets to 200Hz to retain essential information. Meanwhile, we applied a 0.5-70Hz bandpass filter to the EEG and SEEG signals to remove noise, along with a 50Hz notch filter to eliminate power line interference. Finally, z-score normalization is used to reduce nonstationarity and fluctuations in the signals (Hosseini et al., 2020).

**Model Configuration:** Each EEG and SEEG patch has a length of 1 second, with the number of patches limited to 256. The temporal and spatial encoders consist of 3 layers of 1-D convolution, GroupNorm, and GELU, while the frequency encoder is composed of 3 layers of wavelet packet convolution, GroupNorm, and GELU, with the wavelet packet convolution used to extract frequency band features from the signals. Subsequently, the EEG or SEEG patches are transformed into patch embeddings and fed into a Transformer encoder with a depth of 8 layers and 10 attention heads. For more details, please refer to Appendix C.

**Pre-training and Fine-tuning:** To train the epilepsy neural tokenizer and EpilepsyFM pre-training, we utilized the largest publicly available EEG epilepsy TUH series dataset, along with most of the XJSZ-EEG and XJSZ-SEEG datasets as training data. This dataset includes approximately 9,000 patients and 2,000 hours of data, with specific details provided in Appendix B. The task records vary across different datasets. We used TUAB, TUEV, CHB-MIT, and a small portion of XJSZ-EEG and XJSZ-SEEG data from different patients to evaluate downstream epilepsy tasks. The training, validation, and testing sets are divided in a ratio of 80%, 10%, and 10%. Our experiments are conducted on 8 A100-PCIE-40GB GPUs using Python 3.8 (Ubuntu 20.04) and PyTorch 1.11.0 + CUDA 11.3. During training, binary classification tasks employed binary cross-entropy loss, while multi-class classification tasks used cross-entropy loss.

## 3.3 EVALUATION RESULTS

## 3.4 MAIN RESULTS

The performance of EpilepsyFM across all downstream tasks is presented in Figure 4. As a foundational model in the field of epilepsy, EpilepsyFM demonstrates SOTA (state-of-the-art) performance in both extracranial EEG and intracranial SEEG detection and prediction tasks, compared to other baseline models. In the following paragraphs, we will provide a detailed discussion of the experimental results for each task.

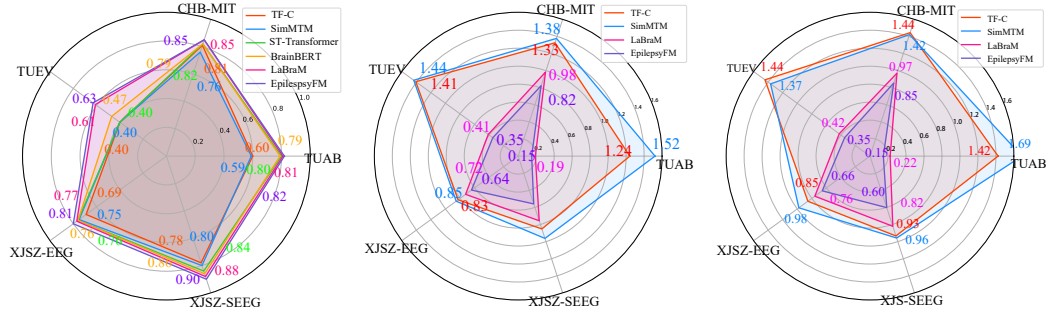

Figure 4: Performance comparison of downstream tasks. **Left:** Epilepsy detection task (Metrics: Balance Accuracy); **Center:** Epilepsy short-term prediction task (Metrics: Mean Absolute Error); **Right:** Epilepsy long-term prediction task (Metrics: Mean Absolute Error).

## 3.5 SEIZURE DETECTION

Epileptic seizure detection is of significant importance in the clinical application of EEG and SEEG (Siddiqui et al., 2020; Tran et al., 2022; Farooq et al., 2023). The aim of seizure detection is to identify or classify the discharge phenomena and types of discharges in the intracranial SEEG and extracranial EEG signals of patients with epilepsy. We present the seizure detection performance of

scalp EEG and intracranial SEEG on different datasets. We use Balance Accuracy, AUC-PR, and AUROC as evaluation metrics, with the definitions of these metrics provided in Appendix D.

**Seizure Detection on Extracranial EEG:** We present the detection performance of EpilepsyFM against other advanced methods on various public epilepsy EEG datasets (TUAB, TUEV, CHB-MIT) and a private EEG dataset (XJSZ-EEG). We compared not only self-supervised pre-training methods (TF-C, SimMTM) (Zhang et al., 2022; Dong et al., 2024) and temporal mask modeling methods (BrainBERT, LaBraM) (Wang et al., 2023; Jiang et al., 2024) as unsupervised approaches but also a supervised learning method specifically for EEG, namely ST-Transformer (Song et al., 2021).

As shown in Table 1, EpilepsyFM outperformed other methods across multiple evaluation metrics, with LaBraM and BrainBERT following closely behind. This is largely attributed to LaBraM and BrainBERT's ability to effectively consider contextual information and integrate embeddings of neural signals related to brain activity, particularly as BrainBERT shares a similar approach to our model in integrating temporal and spectral information.

Table 1: The performance(%) of epilepsy detection tasks on EEG.

| Metrics | TUAB | | | CHB-MIT | | |
|---|---|---|---|---|---|---|
| Methods | Balance Acc. | AUC-PR | AUROC | Balance Acc. | AUC-PR | AUROC |
| TF-C | 59.98±0.47 | 58.08±0.44 | 57.45±0.68 | 80.68±0.66 | 62.74±0.50 | 58.65±0.88 |
| SimMTM | 58.68±0.40 | 59.33±0.44 | 62.15±0.72 | 75.74±0.63 | 58.02±0.81 | 81.38±0.92 |
| ST-Transformer | 79.66±0.23 | 85.21±0.26 | 87.07±0.19 | 82.45±1.27 | 60.32±1.46 | 86.28±0.68 |
| BrainBERT | 78.97±0.75 | 85.14±0.54 | 87.33±0.71 | 82.31±0.72 | 60.45±0.45 | 86.36±0.72 |
| LaBraM | 81.37±0.21 | 89.33±0.24 | 89.72±0.16 | 84.51±0.05 | 62.45±0.73 | 90.47±0.14 |
| **EpilepsyFM** | **82.27±0.18** | **89.77±0.17** | **91.91±0.11** | **85.12±0.34** | **64.25±0.45** | **92.01±0.52** |
| Metrics | XJSZ-EEG | | | TUEV | | |
| Methods | Balance Acc. | AUC-PR | AUROC | Balance Acc. | Cohen Kappa | Weighted F1 |
| TF-C | 68.54±0.43 | 69.35±0.34 | 73.35±0.86 | 40.46±0.67 | 37.19±0.12 | 67.19±0.12 |
| SimMTM | 74.67±0.87 | 75.34±0.56 | 75.85±0.45 | 40.12±0.96 | 37.94±0.73 | 37.94±0.73 |
| ST-Transformer | 76.27±1.64 | 76.79±1.58 | 79.36±0.99 | 39.84±2.28 | 37.65±3.06 | 68.23±1.90 |
| BrainBERT | 75.90±0.36 | 76.45±0.53 | 78.34±0.57 | 47.55±0.75 | 48.76±0.58 | 60.64±0.56 |
| LaBraM | 76.74±0.45 | 77.21±0.84 | 79.98±0.56 | 61.26±0.48 | 61.76±1.21 | 80.46±0.45 |
| **EpilepsyFM** | **80.35±0.83** | **82.05±0.68** | **84.25±0.73** | **62.74±0.76** | **64.35±1.01** | **81.45±0.96** |

**Seizure Detection on Intracranial SEEG:** The detection results for SEEG signals are presented in Table 2. This study found that the detection performance of SEEG signals significantly outperforms that of EEG signals, primarily due to the high signal-to-noise ratio of SEEG signals. This characteristic enables our EpilepsyFM model to excel across multiple evaluation metrics. Additionally, the design of the EpilepsyFM model fully considers the unique properties of SEEG signals, resulting in enhanced robustness across different intracranial channel locations. These findings underscore the critical role of EpilepsyFM in SEEG analysis and further support its effectiveness in clinical applications.

Ultimately, our detection results outperform other methods due to our model's design, which specifically addresses the characteristics of seizure onset and propagation in epilepsy. By utilizing channel

Table 2: The performance(%) of epilepsy detection tasks on SEEG.

| Metrics | XJSZ-SEEG | | |
|---|---|---|---|
| Methods | Balance Acc. | AUC-PR | AUROC |
| TF-C | 78.10±0.50 | 76.50±0.60 | 80.04±0.55 |
| SimMTM | 80.25±0.65 | 78.80±0.70 | 82.85±0.60 |
| ST-Transformer | 83.50±0.70 | 81.56±0.50 | 85.34±0.65 |
| BrainBERT | 86.30±0.60 | 84.50±0.75 | 89.00±0.70 |
| LaBraM | 87.50±0.75 | 83.67±0.61 | 85.46±0.55 |
| EpilepsyFM | **89.74±0.74** | **86.05±0.57** | **92.07±0.56** |

set masking and spatio-temporal frequency encoders, we achieve a broader receptive field. This enables the model to effectively capture complex patterns and relationships within the EEG signals, thereby improving the accuracy of epilepsy detection (further experimental evidence can be found in Appendix G).

## 3.6 EPILEPSY SHORT- AND LONG-TERM SIGNAL PREDICTION

Predicting epilepsy signals is essential for developing early warning systems for patients in need of preventive measures (Usman et al., 2020; Savadkoohi et al., 2020; Dissanayake et al., 2021). We established both short-term and long-term predictions for seizures, focusing on signals of varying durations. Specifically, we set the historical sequence length to 90 epochs (90 seconds), with short-term predictions covering 12 epochs (12 seconds) and long-term predictions spanning 20 epochs (20 seconds). A linear prediction head is utilized to forecast future signals, and we employed Mean Absolute Error (MAE) and Mean Squared Error (MSE) as evaluation metrics, with the definitions of these metrics provided in Appendix D. The prediction results are shown in Table 3.

**Short- and Long-term Signal Prediction on Extracrania EEG**: EpilepsyFM has demonstrated outstanding performance in short-term prediction across multiple EEG datasets. For example, in the XJSZ-EEG dataset, the MAE is 0.6421 and the MSE is 0.9380, which is significantly better than other baseline methods. This superiority is attributed to the model's multi-dimensional design, which enables it to analyze signals across various dimensions, allowing for flexible capture of dynamic changes and long-range dependencies in the signals. In long-term prediction, EpilepsyFM also exhibits strong performance, with a MAE of 0.6567 and an MSE of 1.1123 in the XJSZ-EEG dataset, further validating its potential and reliability in epileptic signal prediction.

**Short- and Long-term Signal Prediction on Intracrania SEEG**: The EpilepsyFM method has again demonstrated its outstanding capabilities in short-term prediction, achieving a MAE of 0.5587 and an MSE of 0.8592 in the XJSZ-SEEG dataset, highlighting its effectiveness in capturing key features in SEEG signal processing. In contrast, the LaBraM and SimMTM methods recorded MAE values of 0.7556 and 0.9621, respectively. While these methods performed well, they did not match the accuracy of EpilepsyFM. In long-term signal prediction, EpilepsyFM continued to maintain its advantage, with a MAE of 0.6024 and an MSE of 0.9502, further emphasizing its adaptability and accuracy across different signal processing scenarios.

Table 3: The performance of epilepsy short- and long-term Signal Prediction.

| Dataset | Methods | Short-term Signal Prediction | | Long-term Signal Prediction | |
| --- | --- | --- | --- | --- | --- |
| | | MAE | MSE | MAE | MSE |
| TUAB | TF-C | 1.2413±0.0013 | 1.8411±0.0037 | 1.4242±0.0012 | 1.9720±0.0021 |
| | SimMTM | 1.5213±0.0015 | 2.2561±0.0063 | 1.6892±0.0010 | 2.4720±0.0023 |
| | LaBraM | 0.1907±0.0025 | **0.3136±0.0081** | 0.2237±0.0030 | 0.3321±0.0013 |
| | EpilepsyFM | **0.1498±0.0010** | 0.3868±0.0076 | **0.1482±0.0026** | **0.2868±0.0016** |
| CHB-MIT | TF-C | 1.3251±0.0011 | 2.2802±0.0053 | 1.4436±0.0017 | 2.7676±0.0011 |
| | SimMTM | 1.3765±0.0014 | 2.3215±0.0062 | 1.4212±0.0012 | 2.8675±0.0010 |
| | LaBraM | 0.9823±0.0013 | 1.4657±0.0076 | 0.9701±0.0014 | 1.5246±0.0018 |
| | EpilepsyFM | **0.8231±0.0014** | **1.2002±0.0048** | **0.8550±0.0012** | **1.2319±0.0016** |
| TUEV | TF-C | 1.4075±0.0008 | 1.6802±0.0110 | 1.4436±0.0016 | 1.6789±0.0031 |
| | SimMTM | 1.4356±0.0024 | 1.7721±0.0143 | 1.3665±0.0033 | 1.6213±0.0030 |
| | LaBraM | 0.4112±0.0023 | 1.6092±0.0064 | 0.4231±0.0012 | 1.6294±0.0013 |
| | EpilepsyFM | **0.3489±0.0046** | **1.5272±0.0044** | **0.3489±0.0014** | **1.5257±0.0043** |
| XJSZ-EEG | TF-C | 0.8265±0.0014 | 1.2588±0.0022 | 0.8541±0.0015 | 1.3187±0.0025 |
| | SimMTM | 0.8531±0.0012 | 1.3065±0.0025 | 0.9779±0.0016 | 1.4231±0.0030 |
| | LaBraM | 0.7231±0.0010 | 1.1214±0.0015 | 0.7583±0.0015 | 1.2780±0.0021 |
| | EpilepsyFM | **0.6421±0.0010** | **0.9380±0.0010** | **0.6567±0.0012** | **1.1123±0.0015** |
| XJSZ-SEEG | TF-C | 0.8513±0.0011 | 1.3034±0.0023 | 0.9332±0.0014 | 1.4015±0.0026 |
| | SimMTM | 0.9621±0.0015 | 1.4295±0.0025 | 0.9577±0.0018 | 1.5389±0.0028 |
| | LaBraM | 0.7556±0.0012 | 1.2674±0.0020 | 0.8206±0.0015 | 1.3423±0.0023 |
| | EpilepsyFM | **0.5587±0.0010** | **0.8592±0.0012** | **0.6024±0.0014** | **0.9502±0.0018** |

In summary, the EpilepsyFM method outperformed other methods in both short-term and long-term prediction tasks on the EEG and SEEG datasets, exhibiting significant accuracy advantages. This result underscores the potential of EpilepsyFM in predicting epilepsy signals.

## 4 EPILEPSYLM WITH/WITHOUT PRE-TRAINING

In the experiments, we compared the impact of pre-training on the detection and prediction performance across downstream datasets. For the without pre-training experiments, we trained the EpilespyFM model from scratch on each downstream task dataset, and the results are shown in Figure 5. Here, we only present the performance of publicly available datasets in the downstream tasks. The experimental results indicate a decline in performance on the TUAB, CHB-MIT, and TUEV datasets, with a particularly noticeable drop on the TUEV dataset, thereby validating the effectiveness of channel set mask pre-training.

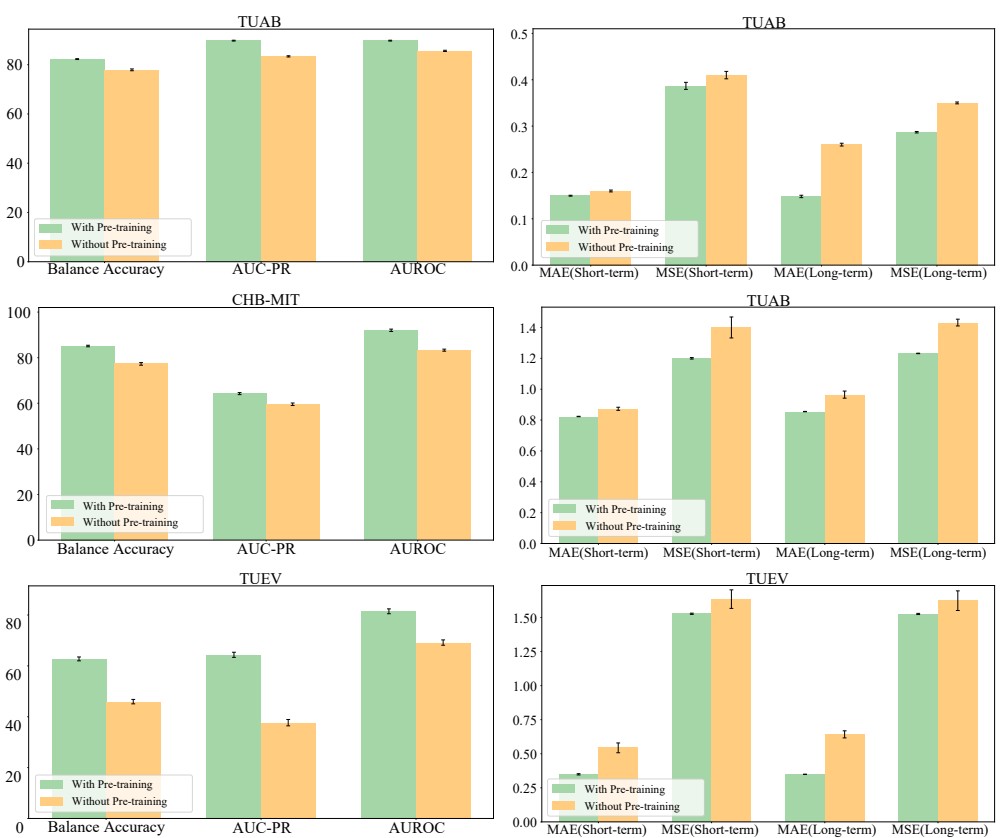

Figure 5: Performance comparison with/without pre-training. **Left:** Epilepsy detection task; **Right:** Epilepsy short- and long-term prediction tasks

.

## 5 CONCLUSION

Inspired by LLMs, we designed a foundational model—EpilepsyFM, specifically for understanding the mechanisms of epilepsy onset and propagation. This model is capable of learning generalized representayions from extracranial EEG and intracranial SEEG signals across different signal types, data formats and sources to facilitate various epilepsy-related tasks. To enhance data diversity and address the scarcity of intracranial SEEG signals, we collected time-locked clinical EEG and SEEG data from a first-class hospital's neurosurgery department. Initially, we trained an epilepsy neural tokenizer to learn comprehensive EEG and SEEG data, thereby creating a codebook for the epilepsy neural tokenizer. We then employed unsupervised pre-training using a channel set masking approach to capture generalized representations of epilepsy. Subsequently, we fine-tuned the model on multiple downstream tasks related to epilepsy, achieving state-of-the-art performance across different task settings. Extensive experiments demonstrated that EpilepsyFM achieved leading performance in tasks such as seizure detection, and short- and long-term neural signal prediction.

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

## A RELATED WORK

**Self-supervised Pre-training:** Self-supervised pre-training has made significant progress in natural language processing and computer vision, yet its potential in the brain-computer interface (BCI) field remains underexplored, providing an opportunity to leverage large-scale EEG data for training large models. Currently, several foundational models have emerged: Neuro-GPT (Cui et al., 2024) aims to address the scarcity and heterogeneity of EEG data by combining EEG encoders with GPT models; MMM (Yi et al., 2023) facilitates cross-dataset brainwave pre-training through a unified topological structure, employing multidimensional position encoding and a multi-level channel hierarchy; LaBraM (Jiang et al., 2024) segments raw EEG signals into slices and trains a neural tokenizer via spectral prediction to generate neural vocabulary, masking portions of slices during pre-training for the Neural Transformer to predict from visible slices. The Brant (Zhang et al., 2023) and Brant-2 (Yuan et al., 2024b) models are specifically designed for intracranial recordings, with Brant-2 expanding upon its predecessor to demonstrate robustness against data variations and modeling scales, applicable to a broader range of neural data and validating the effectiveness and application potential of self-supervised learning in the BCI domain. The introduction of these models marks a significant step toward utilizing self-supervised learning and large-scale datasets in BCI.

**Domain-specific Foundation Model:** Existing brain signal analysis models are primarily general-purpose, excelling in various applications but lacking focus on epilepsy analysis. For instance, domain-specific models have been successfully utilized in other medical fields, such as Tariq et al.'s work on prostate cancer, where they collected over 1.8 million clinical notes and developed a model fine-tuned for clinical information prediction and question-answering tasks. Their model, enhanced by UMLS-guided training and domain-specific tokenizers, significantly outperformed the general-purpose model GPT-2 and even surpassed the specialized BioGPT in certain tasks (Tariq et al., 2024). However, similar methodologies have not been widely applied to epilepsy analysis. While there have been efforts like Zhao et al.'s EpilepsyLLM, which fine-tuned models on epilepsy-related datasets, this work focused mainly on textual knowledge representation (Zhao et al., 2024). In contrast, our research advances a model specifically tailored for epilepsy, leveraging extensive specialized data to analyze seizure dynamics and addressing clinical needs in EEG analysis. Our work not only fills a gap in LLM field but also provides insights for applying large-scale EEG data to specific disease research, further advancing epilepsy studies.

## B DATASET DESCRIPTION

- **TUEP** (Veloso et al., 2017): This corpus is a subset of the TUH EEG Corpus that contains 100 sessions from patients with epilepsy and 100 sessions from patients without epilepsy with EEG recorded (19-23 channels, 256 Hz). These data are based on clinical history, EEG features, etc., reviewed and verified by a board-certified neurologist. (total time: 473 hours)

- **TUSL** (von Weltin et al., 2017): This is another subset of TUH EEG that contains sessions that are known to contain seizure events, slowing events, and complex background events (23 channels, 256 Hz). The corpus has been used in studies to distinguish between seizures and slowing events. (total time: 20.59 hours)

- **TUSZ** (Shah et al., 2018): This corpus contains EEG signals that are annotated to contain in addition to a start and stop time, a channel index with EEG recorded (19-23 channels, 256 Hz). (total time: 1138.53 hours)

- **TUAB** (Obeid & Picone, 2016): The corpus contains statistical data (about demographics and related information) from the TUH EEG anomaly Corpus, which contains EEG recordings is classified as clinically normal or abnormal. (total time: 1142 hours)

- **TUEV** (Obeid & Picone, 2016): This is a subset of TUH EEG that contains annotations of EEG segments as one of six classes: (1) spike and sharp wave (SPSW), (2) generalized periodic epileptiform discharges (GPED), (3) periodic lateralized epileptiform discharges (PLED), (4) eye movement (EYEM), (5) artifact (ARTF) and (6) background (BCKG).

- **CHB-MIT** (Shoeb, 2009): This dataset consists of scalp EEG signals from 23 pediatric epilepsy patients, including 24 EEG sets, each subset corresponding to several days of records from one epilepsy patient, and each epileptic EEG set contains 9 to 42 consecutive sets of multilead epileptic EEGs (256 Hz) (total time: 969hours)

- **XJSZ-EEG:** The dataset is collected from the Department of Neurosurgery at a first-class hospital, monitoring the EEG signals of 12 clinical epilepsy patients. Each patient have different electrode positions and channel numbers. We selected the data from 4 patients for self-supervised pre-training and used the data from 8 patients for downstream tasks. The labels are manually annotated by two clinically experienced physicians. The sampling frequency is 1000 Hz, synchronized with SEEG signals. We chose seizure subfolders (6 hours per subfolder) for training, with a total duration of approximately 38 hours, as shown in Table 4.

Table 4: General information of epileptic patients of the XJSZ-EEG dataset.

| Patient ID | Gender | Age | EEG Channels | Seizure Events | Total Seizure Time (s) | Total Time (h) |
|---|---|---|---|---|---|---|
| 01 | Male | 24 | 19 | 14 | 420 | 98.3 |
| 02 | Male | 16 | 19 | 13 | 390 | 125.5 |
| 03 | Male | 34 | 18 | 32 | 960 | 100.5 |
| 04 | Male | 14 | 19 | 18 | 540 | 244.7 |
| 05 | Male | 16 | 19 | 13 | 390 | 125.5 |
| 06 | Male | 13 | 18 | 32 | 962 | 100.5 |
| 07 | Male | 14 | 19 | 18 | 540 | 244.7 |
| 08 | Male | 8 | 18 | 41 | 1231 | 401.5 |
| 09 | Female | 17 | 18 | 6 | 362 | 24 |
| 10 | Female | 16 | 18 | 32 | 960 | 237 |
| 11 | Female | 5 | 17 | 43 | 1295 | 166 |
| 12 | Male | 34 | 19 | 3 | 107 | 72 |

- **XJSZ-SEEG:** The dataset is collected from the Department of Neurosurgery at a first-class hospital, monitoring the SEEG signals of 20 clinical epilepsy patients. All signals are recorded using the Neruacle digital SEEG machines (NSH0256). Each patient have a personalized surgical plan, and the number of SEEG contacts varied, with each contact representing an SEEG channel. The dataset is manually annotated by two clinically experienced physicians. The sampling frequency is 1000 Hz, with a total duration of 388 hours. We used the data from 12 patients for pre-training, not by summing the total duration of each patient's data but by selecting seizure subfolder data from each patient. The remaining 8 patients are used for downstream tasks, as shown in Table 5.

Table 5: General information of epileptic patients of the XJSZ-SEEG dataset.

| Patient ID | Gender | Age | SEEG Electrodes | SEEG Channels | Seizure Events | Total Seizure Time (s) | Total Time (h) |
|---|---|---|---|---|---|---|---|
| 01 | Female | 33 | 9 | 127 | 146 | 4380 | 384 |
| 02 | Female | 5 | 10 | 132 | 65 | 10549 | 154 |
| 03 | Female | 11 | 8 | 126 | 138 | 2438 | 94 |
| 04 | Male | 15 | 13 | 196 | 12 | 2430 | 237 |
| 05 | Female | 10 | 11 | 142 | 16 | 1067 | 92 |
| 06 | Male | 28 | 13 | 166 | 26 | 1222 | 141 |
| 07 | Male | 7 | 10 | 130 | 17 | 1020 | 169 |
| 08 | Female | 17 | 11 | 141 | 6 | 362 | 24 |
| 09 | Female | 34 | 14 | 190 | 5 | 202 | 120 |
| 10 | Female | 11 | 10 | 141 | 14 | 427 | 67 |
| 11 | Female | 28 | 10 | 141 | 11 | 385 | 102 |
| 12 | Male | 34 | 13 | 200 | 3 | 107 | 72 |
| 13 | Male | 16 | 13 | 166 | 13 | 390 | 125.5 |
| 14 | Male | 14 | 13 | 188 | 18 | 540 | 244.7 |
| 15 | Male | 8 | 9 | 106 | 41 | 1231 | 401.5 |
| 16 | Male | 34 | 14 | 194 | 32 | 962 | 100.5 |
| 17 | Female | 5 | 10 | 120 | 43 | 1295 | 166 |
| 18 | Female | 11 | 10 | 118 | 138 | 4142 | 93.5 |
| 19 | Female | 34 | 11 | 127 | 44 | 1326 | 118.5 |
| 20 | Male | 24 | 13 | 195 | 5 | 157 | 98.3 |

# C MODEL DETAILS

The EpilepsyFM Tokenizer Training model comprises three key components: (1) the Epilepsy Neural Tokenizer, (2) the Vector Quantizer, and (3) the EpilepsyFM Neural Decoder, as shown in Table 6. As illustrated in Figure 2, the architecture of the Neural Tokenizer efficiently processes and tokenizes epilepsy-related data. The Vector Quantizer is designed similarly to the LaBraM model, ensuring effective data compression and representation. The EpilepsyFM Neural Decoder features a Transformer decoder with stacked layers and a temporal regression head, enabling accurate modeling of time series data. Additionally, the pre-training of EpilepsyFM aligns with the Epilepsy Neural Transformer module, as shown in Table 7.

Table 6: The hyperparameters for EpilepsyFM Tokenizer Training.

| Module | Sub-Module | Name | Value |
|---|---|---|---|
| Epilepsy Neural Tokenizer (Epilepsy Neural Transformer) | Temporal Encoder | # of Input Channels | {1,8,8} |
| | | # of Output Channels | {8,8,8} |
| | Spectral Encoder | # of Input Channels | {1,2,2} |
| | | # of Output Channels | {2,2,2} |
| | Spatial Encoder | # of Input Channels | {10,16,16} |
| | | # of Output Channels | {16,16,8} |
| | Transformer Encoder | # of Transformer Layers | 8 |
| | | Hidden Size | 200 |
| | | MLP Size | 800 |
| | | # of Attention Heads | 10 |
| Vector Quantizer | - | Codex Size | 8192×64 |
| | | Embedding-to-Codex | $200 \rightarrow 200(Tanh) \rightarrow 64$ |
| | | Codex-to-Embedding | $64 \rightarrow 200$ |
| Epilepsy Neural Decoder | Transformer Decoder | # of Transformer Layers | 3 |
| | | Hidden Size | 200 |
| | | MLP Size | 800 |
| | | # of Attention Heads | 10 |
| Optimizer | - | Batch Size | 512 |
| | | Maximum Learning Rate | 5e-5 |
| | | Minimum Learning Rate | 1e-6 |
| | | Learning Rate Scheduler | Cosine |
| | | Optimizer Type | AdamW |
| | | Adam | (0.9,0.99) |
| | | Weight Decay | 0.01 |
| | | Total Epochs | 100 |
| | | Warm-up Epochs | 30 |

Table 7: The hyperparameters for the EpilepsyFM.

| Hyperparameters | Values |
|---|---|
| Batch Size | 64 |
| Maximum Learning Rate | 5e-4 |
| Minimal Learning Rate | 1e-6 |
| Learning Rate Scheduler | cosine |
| Optimizer | adamw |
| Adam $\beta$ | (0.9, 0.99) |
| Weight Decay | 0.05 |
| Total Epochs | 30 |
| Warmup Epochs | 5 |
| Drop Path | 0.1 |
| Layer-wise Learning Rate Decay | 0.9 |
| Label Smoothing | 0.1 |

## D    EVALUATION METRICS

In the task of epileptic seizure detection(Cai et al., 2023; Chen et al., 2022), we follow existing research and employ balanced accuracy, AUC-PR, and AUROC as evaluation metrics for binary classification tasks. For multi-class classification tasks(Memar & Faradji, 2017), we use balanced accuracy, Cohen Kappa, and weighted F1 as evaluation metrics. In the short-term and long-term prediction tasks for epilepsy (Woo et al., 2022), we also adhere to existing research by utilizing MAE, MSE as evaluation metrics.

- **Balanced accuracy:** Balanced accuracy is a method of calculating accuracy that accounts for class imbalance. It is the average of the recall for each class. In epilepsy detection, balanced accuracy effectively reflects the model's classification performance in the face of imbalanced data, avoiding the bias that can arise from simple accuracy.

- **AUC-PR**: AUC-PR measures the area under the curve plotted between precision and recall, primarily used to evaluate the performance of classification models on imbalanced data. In epilepsy datasets, AUC-PR better assesses model performance in scenarios with scarce positive samples, particularly focusing on the precision and recall of the model.

- **AUROC:** AUROC measures the area under the receiver operating characteristic curve, primarily used to evaluate the performance of classification models across various threshold levels. In epilepsy datasets, AUROC better assesses the overall performance of the model under different thresholds.

- **Cohen Kappa:** Cohen Kappa is a statistic that measures the agreement between two classifiers while accounting for the effect of random chance. Its value ranges from -1 to 1, where 1 indicates perfect agreement, 0 indicates no better than random agreement, and negative values indicate agreement below random levels. In multi-class tasks, Cohen Kappa effectively evaluates the consistency of the model's classifications, especially in situations of class imbalance.

- **Weighted F1 Score:** Weighted F1 Score is an extension of the F1 score that considers the importance of each class. It calculates the F1 score for each class and weighs it according to the frequency of that class in the dataset. The Weighted F1 Score is particularly important in multi-class tasks, as it balances the performance of all classes, making it suitable for addressing class imbalance issues and ensuring effective evaluation of the model across all categories.

- **Mean Absolute Error (MAE):** MAE calculates the average of the absolute differences between predicted values and actual values. MAE treats all errors equally, avoiding the bias introduced by MSE towards extreme values, thus providing an intuitive assessment of model performance. In epilepsy prediction, MAE quantifies the overall predictive accuracy of the model clearly, particularly when ensuring the reliability of each prediction is critical.

- **Mean Squared Error (MSE):** MSE refers to the average of the squared differences between predicted values and actual values. The primary advantage of MSE lies in its sensitivity to larger errors, making it useful in model optimization to identify and reduce significant biases. In the context of epilepsy prediction, MSE effectively reflects how well the model captures rare but severe seizure events.

## E    VISUALIZATION OF MASK MODELING

In Figure 6, we present the convergence curves of the total pre-training loss and the masked modeling accuracy of EpliespyFM. It is evident from the figure that the training method utilizing channel set masked modeling effectively facilitates the learning process of the model. Specifically, the training loss steadily decreases with each iteration, indicating that the model is progressively optimizing its parameters to enhance performance. At the same time, the masked accuracy shows a consistent upward trend, further validating the effectiveness of this approach in capturing the features of EEG and SEEG data.

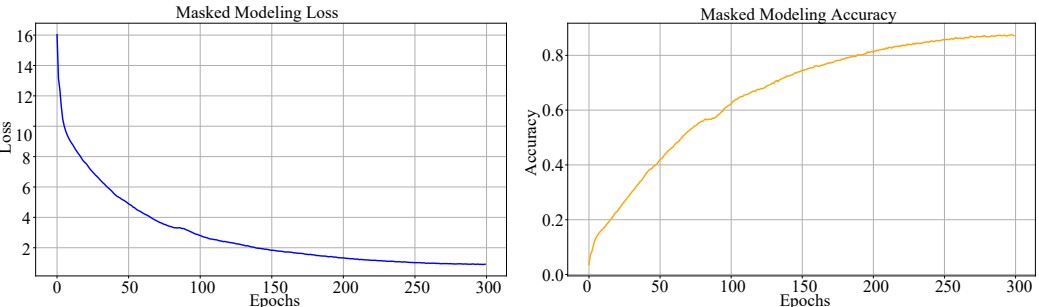

Figure 6: The loss curve and accuracy curve during the training process of the EpliespyFM.

## F ABLATION ON CHANNEL-SET MASKING

Due to the locality and specificity of brain computation, research into more challenging epilepsy tasks (such as the clustering discharges and propagation circuits of different brain regions in different patients) remains to be explored. Our designed channel masking strategy can cover the neural activity signals of multiple neighboring channels, which exhibit signal similarity, as shown in Figure 7. SEEG signal channels are continuous, so there is no need to consider channel reordering. For EEG signals, we reorder the electrode signals based on the left and right hemispheres, grouping nearby brain region channels together to implement the masking.

In the experiments, we evaluated the performance of EEG seizure detection using different numbers of masked channels, as shown in Table 8. From the figure, it can be observed that in most datasets, the detection performance is best with three channels, and the masking methods using three or four channels perform slightly better than the single-channel masking approach. Utilizing multiple channels for masking can more effectively capture and analyze the characteristic information of seizure events. Compared to single-channel masking, this approach may provide richer spatial and temporal information, enhancing the model's ability to learn complex signal patterns.

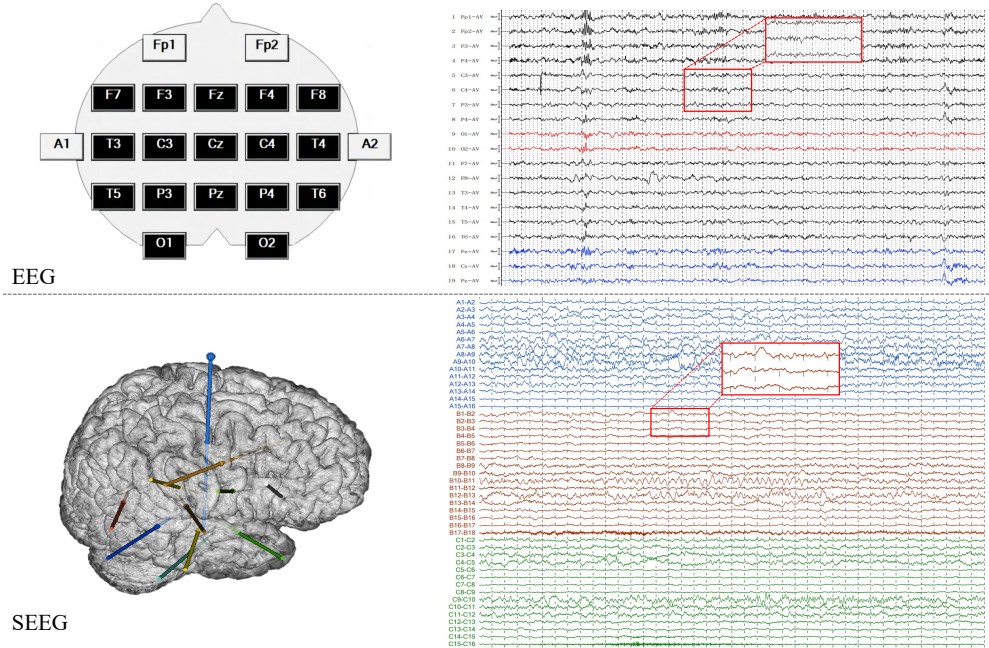

Figure 7: The layouts and information of EEG and SEEG electrode channels.

Table 8: The ablation study of channel set masking.

| Channel-set Mask | TUAB | | | CHB-MIT | | |
|---|---|---|---|---|---|---|
| | Balance Acc. | AUC-PR | AUROC | Balance Acc. | AUC-PR | AUROC |
| 1 | 81.77±0.54 | 86.73±0.37 | 89.53±0.56 | 83.45±0.07 | 61.98±0.35 | **93.32±0.15** |
| 3 | **82.27±0.18** | **89.77±0.17** | **91.91±0.11** | 85.12±0.34 | **64.25±0.45** | 92.01±0.52 |
| 4 | 82.10±0.27 | 89.66±0.12 | 90.91±0.21 | **85.42±0.51** | 64.00±0.57 | 91.41±0.47 |

| Channel-set Mask | XJSZ-EEG | | | TUEV | | |
|---|---|---|---|---|---|---|
| | Balance Acc. | AUC-PR | AUROC | Balance Acc. | AUC-PR | AUROC |
| 1 | 79.35±0.87 | 81.58±0.67 | **84.69±0.48** | 61.76±0.58 | 61.33±0.65 | 80.34±0.68 |
| 3 | 80.35±0.83 | **82.05±0.68** | 84.25±0.73 | **62.74±0.76** | 64.35±1.01 | 81.45±0.96 |
| 4 | **81.57±0.26** | 81.45±0.73 | 86.37±0.36 | 62.25±0.38 | **67.11±0.46** | **81.79±0.47** |

# G  ABLATION ON ENCODER MODULE

In the experiments, we found that the temporal encoder (TEME), spectral encoder (SPEE), and spatial encoder (SPAE) significantly impact the performance of the epilepsy model, as shown in Table 9. Removing the temporal encoder leads to a noticeable decline in balanced accuracy and AUC metrics, indicating that temporal information is crucial for capturing the dynamic characteristics of epileptic seizures. The absence of the spectral encoder also affects predictive performance, particularly in the analysis of long time series. Although the impact of the spatial encoder is relatively smaller, its removal still results in a performance drop, highlighting the importance of spatial features. Ultimately, the EpilepsyFM model, which combines all encoder modules, demonstrates the best performance.

Table 9: The ablation study of encoder module

| - | TUAB | | | CHB-MIT | | |
|---|---|---|---|---|---|---|
| | Balance Acc. | AUC-PR | AUROC | Balance Acc. | AUC-PR | AUROC |
| w/o TEME | 81.47±0.15 | 89.32±0.10 | 89.88±0.08 | 82.12±0.20 | **65.23±0.15** | 90.12±0.12 |
| w/o SPEE | 80.45±0.18 | 88.75±0.12 | 89.00±0.09 | 81.50±0.25 | 64.10±0.20 | 89.50±0.15 |
| w/o SPAE | 79.65±0.20 | 87.40±0.15 | 88.50±0.10 | 80.00±0.30 | 63.55±0.25 | 88.80±0.20 |
| EpilepsyFM | **82.27±0.18** | **89.77±0.17** | **91.91±0.11** | **85.12±0.34** | 64.25±0.45 | **92.01±0.52** |

| - | XJSZ-EEG | | | TUEV | | |
|---|---|---|---|---|---|---|
| | Balance Acc. | AUC-PR | AUROC | Balance Acc. | AUC-PR | AUROC |
| w/o TEME | 80.13±0.23 | 81.86±0.43 | 83.32±0.66 | **62.76±0.33** | 61.76±1.23 | 80.45±0.45 |
| w/o SPEE | 79.00±0.51 | 81.20±0.34 | 83.00±0.41 | 61.50±0.60 | 63.00±1.00 | 80.20±0.90 |
| w/o SPAE | 78.50±0.43 | 80.00±0.25 | 82.00±0.32 | 60.80±0.55 | 62.50±0.80 | 79.80±0.75 |
| EpilepsyFM | **80.35±0.83** | **82.05±0.68** | **84.25±0.73** | 62.74±0.76 | **64.35±1.01** | **81.45±0.96** |

# H  DISCUSSION

**Limitations.** Although EpilspyFM has made significant progress in common clinical tasks for epilepsy by training on multiple publicly available EEG datasets and private data, the availability of SEEG data in clinical settings remains relatively limited. This somewhat constrains our in-depth exploration in the field of foundational models for epilepsy. If we can further enrich task settings, particularly in the precise localization of epileptic foci and avoidance of functional areas, it will greatly enhance the clinical applicability of the model, allowing for more effective guidance in treatments such as thermal coagulation.

**Outlook.** Based on the aforementioned limitations, our research outlook encompasses several key areas: First, we plan to expand the collection of SEEG data, particularly from different devices, to enhance the diversity and representativeness of the data; we are currently actively promoting

this effort. Second, we will extend the task settings to consider the relationship between epileptic lesion areas and brain functional regions, thereby improving the comprehensiveness and depth of our research. Lastly, we will explore the potential of multimodal integration. Our existing model already includes EEG and SEEG as two types of neural signals, and in the future, we may consider incorporating fMRI or PET data. This direction not only poses technical challenges but also represents an innovative approach with significant clinical implications. Through these efforts, we aim to achieve a deeper understanding and breakthroughs in the field of epilepsy research.

