# OpenReview forum: "EpilepsyFM: Foundation Model for Learning Generalized Epileptic Representations from EEG and SEEG Signals"
_ICLR.cc/2025/Conference — Submitted to ICLR 2025_

### Official Review · Reviewer_ACSb · 2024-10-25

**Soundness:** 3
**Presentation:** 2
**Contribution:** 2
**Rating:** 3
**Confidence:** 5

**Summary:**

This paper proposes a foundation model for generalized epileptic representations from EEGs or SEEGs. Using a VQ-VAE-based tokenizer, wavelet packet convolution, and masking-based pertaining, the proposed method can effectively handle both EEG and SEEG signals. Numerous experiments demonstrated the validity of the proposed approach.

**Strengths:**

- The masking strategy, which considers the characteristics of EEG and SEEG, is simple yet more effective for the targeted domain than conventional masking techniques.
- The authors have comprehensively assessed the quality of EpilepsyFM using a wide range of public datasets and various types of epilepsy-related downstream tasks.

**Weaknesses:**

- Regarding the challenges, especially 2 and 3 on Page 2, this reviewer cannot find methodological solutions in the proposed framework to tackle them.
- The explanation for why EEG and SEEG data are processed simultaneously in a single model is insufficient. No reference or experiment demonstrates that using both EEG and SEEG together helps the Epilepsy-specific foundation model learn better representations, making the rationale for using both datasets unconvincing.
- Additionally, the method lacks techniques for handling the differences in EEG and SEEG data characteristics when processing them in a single model. EEG and SEEG are different types of signals with several distinctions, such as measurement locations and commonly used sampling rates (SEEG contains more high-frequency components than EEG [1]). These differences in signal characteristics should be considered during the training process. However, the proposed method does not adequately address this aspect.
- Furthermore, the chosen baselines are insufficient to demonstrate the proposed method's effectiveness fully. Specifically, the baseline LaBraM uses only EEG data, while BrainBERT uses only SEEG data. Therefore, a valid rationale for using each model as a baseline for tasks involving the other signal type is required, or an explanation of whether more suitable baselines exist should be provided.
- While the authors presented their experimental results for justification, the 3-channel group masking needs a more rigorous description or mechanistic explanation. Note that the results in Table 8 seem to have no significant difference among channel-number groupings.
- From a technical perspective, this reviewer finds no meaningful contribution to this work. Each module is taken from the previous work with no apparent justification. The experimental results show no significant difference, especially compared to LaBraM. Their performance difference on SEEG seems to result from using SEEG during training, i.e., an imbalance in the training datasets between models.

**Questions:**

- Eq. (2) needs checking. It is unclear and hard to understand. What is $e_{i}^{f}$, and how is it obtained?
- Wavelet packet convolution is introduced by Wang et al., 2021. Then, what do the authors mean to propose in this work? Proposed to use it? It needs to be clarified.
- What is the purpose of Eq. (3), and where are the values of $y_A(t)$ and $y_D(t)$ used?

---

### Official Review · Reviewer_xcrd · 2024-11-01

**Soundness:** 3
**Presentation:** 2
**Contribution:** 2
**Rating:** 5
**Confidence:** 3

**Summary:**

This paper introduces EpilepsyFM, a foundational model specifically designed for epilepsy diagnosis, capable of learning generalized epileptic representations from both EEG and SEEG signals. EpilepsyFM addresses significant limitations in existing deep learning models related to generalization and perception capabilities. Through unsupervised pre-training across various signal types, data formats, and sources, EpilepsyFM demonstrates state-of-the-art performance in seizure detection and signal prediction tasks, showcasing strong generalization potential and promising clinical application.

**Strengths:**

1. EpilepsyFM presents a multi-dimensional encoding design that greatly enhances feature extraction for epilepsy-related tasks.

2. By integrating clinical SEEG data and public EEG data, the model demonstrates strong generalization potential and applicability in real-world settings.

3. The model design effectively incorporates the propagation characteristics of epilepsy, showing somehow the interpretable understanding of this specific domain.

**Weaknesses:**

1. One of the main advantages of foundation models is their ability to perform well across multiple tasks, which seems to be underexplored in this paper. This raises some confusion about whether the motivation for developing a foundation model specifically for a single epilepsy-related scenario is sufficiently compelling. Could the authors clarify the rationale behind applying a foundation model in such a specialized setting？

2. While the model’s interpretability is briefly mentioned, a more in-depth analysis of the learned representations, particularly in relation to clinical insights, would significantly enhance its practical relevance. Expanding on interpretability could also increase the model’s value for clinical applications.

3. The paper lacks a discussion on the computational demands of EpilepsyFM, especially given the model's inherent complexity. Providing details on efficiency and possible optimization strategies would be valuable for assessing its practicality in clinical environments.

I will reconsider my assessment upon reviewing the rebuttal.

**Questions:**

Plz go and check weaknesses

---

### Official Review · Reviewer_AAwo · 2024-11-01

**Soundness:** 2
**Presentation:** 3
**Contribution:** 2
**Rating:** 3
**Confidence:** 4

**Summary:**

This paper proposes a foundation model for accurate seizure detection using sEEG. The proposed method extracts features from time, frequency, and spatial domains, and extensive experiments show that the method achieves state-of-the-art results across different tasks, including seizure detection and both short-term and long-term predictions.

**Strengths:**

1) iEEG is important for the clinical diagnosis and detection of seizures. Focusing on iEEG has significant clinical value.
2) Using LLMs is interesting, as a general representation can serve various downstream tasks.

**Weaknesses:**

1) The proposal, including the VQ-tokenizer, the reconstruction loss in the first training phase, and mask prediction, is quite similar to LaBraM. What are the technical contributions of EpilepsyFM?
2) iEEG does not follow the standard 10-20 system, as it focuses on recording from specific, localized brain areas. How can such spatial information be generalized to different channels or brain regions?
3) The datasets used in this paper are re-extracted from TUH, which may not be sufficient to support a large model. I do not agree that using this dataset alone constitutes a significant contribution.
4) Authors claim to address the lack of consideration for seizure mechanisms by proposing channel set masking. While localizing the seizure onset zone might serve as a more effective marker, early abnormal electrical changes (patterns) are also very important for detection and contribute to the channel issue. I do not believe that simply applying masking can adequately investigate seizure mechanisms.
5) Why does pre-training not work well for predictions in Figure 5?

**Questions:**

Please refer to weakness.

---

### Official Review · Reviewer_3qJK · 2024-11-02

**Soundness:** 2
**Presentation:** 2
**Contribution:** 3
**Rating:** 3
**Confidence:** 5

**Summary:**

The paper proposes EpilepsyFM, a foundational model for epileptic EEG and SEEG.
The EEGs are divided into multiple segments, and a discrete
EEG and SEEG neural tokenizer is trained leading to a domain-specific neural
codebook for epilepsy. A channel set masking
strategy is applied to enhance the model’s ability to capture the spatiotemporal characteristics
of the signals. Experiments are conducted to assess the perfomance of EpilepsyFM  in a variety
of domain-specific tasks, including seizure detection and both short-term and
long-term seizure prediction.

**Strengths:**

Foundational models for EEG, and in particular seizure EEG, are very much needed. Indeed EEGs recorded at different institutions with different machines look different, leading to poor generalization. A foundation model, trained on numerous large datasets, may yield better generalization, an important step towards practical implementation in hospitals.

**Weaknesses:**

First of all, the writing of the paper should be improved. There are too many grammatical errors, e.g., in "For epilepsy, a clinical neurological disorder, we hope to break through the problem that deep learning can only solve a single task, ...."

Moreover, I doubt that the paper has been checked by an epileptologist, since there are statements that are simply incorrect. For instance: "Extracranial electroencephalography (EEG) and intracranial stereoelectroencephalography (SEEG) are crucial for epilepsy diagnosis." SEEG is not used for diagnosis but for surgical planning instead. SEEG requires implanting electrodes inside the brain, and is only done if the patient has been diagnosed with epilepsy. Another example: "Predicting epilepsy signals is essential for developing early warning systems for patients in need of preventive measures". The authors probably mean the prediction of seizures instead of "epilepsy signals".

Various foundation models for EEG have been proposed already, as mentioned in the paper. Moreover, recently a different foundation model for epileptic EEG has been proposed (mot mentioned in the paper): "Nested Deep Learning Model Towards a Foundation Model for Brain Signal Data", https://arxiv.org/html/2410.03191v2. The novelty of this submission is limited, as similar models have been proposed before. The novelty seems to lie in the fact that the model has been trained on EEG and SEEG. However, I dont understand the benefit of training a single model for EEG and SEEG, as both types of recordings are very different in terms of recording procedure and signal characteristics.

**Questions:**

The authors dont clearly describe whether they are detecting seizure EEG segments, or whether they are determining the start and end times of seizures. Only the latter is clinically relevant, while the former is the first step towards the latter.

How is seizure prediction defined precisely? What is a true vs. false positive/negative?

---

### Official Review · Reviewer_M4Xy · 2024-11-04

**Soundness:** 3
**Presentation:** 2
**Contribution:** 2
**Rating:** 6
**Confidence:** 4

**Summary:**

The paper proposes EpilepsyFM, a foundational model tailored specifically for epilepsy-related tasks, such as seizure detection and prediction, using EEG and SEEG signals. The model’s goal is to address several limitations in current deep learning approaches, such as generalization challenges and limited data diversity, especially with SEEG signals. The authors aim to provide a domain-specific model that incorporates seizure onset and propagation mechanisms to enhance clinical applicability. The model achieves SOTA performance on seizure detection tasks across several datasets, both with EEG and SEEG.

**Strengths:**

The paper addresses a need in epilepsy research by proposing a foundational model, EpilepsyFM, specifically tailored for epilepsy-related tasks using EEG and SEEG signals. This focus enhances its potential impact on clinical applications. By including intracranial SEEG data—which is less commonly utilized due to its scarcity—the model has the potential to offer deeper insights, given SEEG's higher signal-to-noise ratio compared to scalp EEG. Introducing a channel set masking strategy is a clever approach to mimic the clustered neuronal discharges characteristic of epilepsy. This method aims to improve the model's ability to capture spatiotemporal features of epileptic activity. Using temporal, spectral, and spatial encoders allows the model to capture complex signal dynamics, which is important for modeling the propagation mechanisms of seizures. The model is evaluated on multiple datasets and tasks, including seizure detection and both short-term and long-term signal prediction, demonstrating its versatility across different scenarios.

**Weaknesses:**

The model appears to be an extension of LABRAM without significant novel contributions. The improvements over LABRAM are marginal on key tasks, which raises concerns about the originality and necessity of the proposed approach. The metrics used (MAE,MSE) may not fully capture the clinical relevance of seizure prediction performance. While reconstructing the signal is useful, much of the problem lies in detecting the seizures, and the length of the seizures. The paper provides insufficient detail on the channel set masking approach. Specifically, the rationale for grouping channels into sets of three and how this impacts the model's learning is not well-explained, making it difficult to understand or reproduce the method. The channel masking strategy seems to focus on local channel clusters. Still, it does not account for long-range connections between brain regions, which are particularly important in SEEG data due to its extensive coverage of intracranial areas.

**Questions:**

Could you elaborate on the channel set masking strategy? Specifically, why did you choose to group channels in sets of three, and how does this choice reflect the physiological characteristics of epileptic discharges?

How does your model account for long-distance interactions between channels, especially in SEEG data where such connections are significant? Have you considered incorporating global spatial relationships into the model?

Have you considered evaluating the model on seizure prediction with more predictive metrics?

SEEG data is inherently scarce. How did you address potential overfitting or bias due to the limited SEEG dataset? Have you tested the model's generalizability across different patient cohorts or institutions?

---

### Meta-Review · Area_Chair_MYKh · 2024-12-13

**Metareview:**

This work proposes a pretraining strategy for EEG and iEEG data in the context of epilepsy. Novelty is considered limited by reviewers, and the rationale for combining EEG and iEEG in one model is not fully clear. Authors did not engage with all reviewers who overall are not convinced by the proposed contribution.

**Additional Comments On Reviewer Discussion:**

Authors did not engage with all reviewers who overall are not convinced by the proposed contribution.

---

### Decision · Program_Chairs · 2025-01-22

Reject